# Usability of machine learning algorithms based on electronic health records for the prediction of acute kidney injury and transition to acute kidney disease: A proof of concept study

Lorenzo Ruinelli [1,2], Pietro Cippà [3,4], Chantal Sieber[3], Clelia Di Serio[2], Paolo Ferrari[5,6], Antonio Bellasi [3,4]*

**1** Area ICT, Ente Ospedaliero Cantonale, Bellinzona, Switzerland, **2** Clinical Trial Unit, Ente Ospedaliero Cantonale, Bellinzona, Switzerland, **3** Division of Nephrology, Ente Ospedaliero Cantonale, Lugano, Switzerland, **4** Faculty of Biomedical Sciences, Università della Svizzera italiana, Lugano, Switzerland, **5** Department of Medicine, Ente Ospedaliero Cantonale, Bellinzona, Switzerland, **6** Clinical School, University of New South Wales, Sydney, Australia

* antonio.bellasi@eoc.ch, antonio.bellasi@usi.ch

## Abstract

### Background

Acute kidney injury (AKI) and acute kidney disease (AKD) are frequent complications of hospitalization, resulting in reduced outcomes and increased cost burden. However, these conditions are only sometimes recognized and promptly treated. Leveraging electronic health records (EHR), we explored the potential of artificial intelligence (AI) in diagnosing AKI and AKD during hospitalization.

### Methods

We retrospectively analyzed EHRs collected from all patients admitted in 2022 to our public hospital network. AKI and AKD were defined according to international guidelines. The database was divided into training and validation sets. Machine Learning (ML) algorithms were developed with 10-fold cross-validation, and diagnostic accuracy was evaluated.

### Findings

We analyzed 34,579 hospitalizations (mean age of 60 years, 50% females). Baseline renal function was available in ~50% of cases. AKI and AKD complicated 10% and 1.5% of hospitalizations, respectively. The majority of AKI episodes (77%) occurred within the first three days of hospitalization, and >50% of subjects with AKI were discharged before complete renal function recovery. ML accurately predicted AKI (AUC-ROC 79%) during hospitalization, based on data available before and at hospital admission. Among subjects with AKI on the first day and longer in-hospital

**Data availability statement:** The data underlying this study cannot be shared publicly due to current data protection regulations, as they contain potentially identifying or sensitive patient information. Access to the data may be granted to researchers who meet the criteria for access to confidential data, subject to approval by the relevant Ethics Committee and the Data Protection Officer (DPO). Data requests must be submitted to the Clinical Trial Unit of the Ente Ospedaliero Cantonale (EOC) at the following contact: Clinical Trial Unit dell'Ente Ospedaliero, Lugano, Via G. Buffi 13, 6900 Lugano; email: clinicaltrialunit@eoc.ch.

**Funding:** The author(s) received no specific funding for this work.

**Competing interests:** The authors have declared that no competing interests exist.

observation, the ML accuracy in predicting AKD transition increased (AUC-ROC from 76% to 88%) by integrating EHR accumulated during the hospitalization. The negative predictive value (NPV) progressively increased from 94% to 98% consistently. Shapely additive explanations documented that age, urgent hospital admission, AKI severity, and baseline renal function were associated with AKI. Renal function trajectory during the first days of hospitalization was the most relevant predictor of AKD.

## Interpretation

ML, relying on EHR before and during hospitalization, may accurately predict AKI and AKD. The high NPV also suggests its implementation as a tool to rule out the risk of renal failure, aid in individualizing patient care, and allocate healthcare resources.

---

## Introduction

Acute kidney injury (AKI), defined as a rapid reduction in renal function according to KDIGO criteria, is a common clinical condition, particularly in hospitalized patients [1–3]. While a large proportion of subjects with AKI experiences a complete renal function recovery, some patients develop a substantial and irreversible nephron loss, leading to acute kidney disease (AKD – the persistence of kidney dysfunction for more than seven days), which can lead to chronic kidney disease (CKD – the persistence of kidney disease for more than 90 days) and progress to kidney failure [1–4].

AKI and AKD are associated with significant morbidity and mortality both during hospitalization and after discharge, particularly among elderly subjects [5–7], regardless of the cause of AKI [1,8]. Impaired renal function is also associated with prolonged hospitalizations and a significant increase in treatment-related costs [9]. In the United States, it is estimated that hospital-acquired AKI affects approximately 498,000 patients, with annual expenses between $4.7 and $24 billion [9]. Despite the clinical and economic relevance of AKI and AKD, an underdiagnosis and under coding [10] have been reported, likely due to the lack of specific symptoms, the overlay/combination with other diagnoses, the complexity of following renal function trajectories and the source data (administrative vs laboratory data) used to define AKI [11].

Prevention and early detection of AKI and AKD should be prioritized to improve patients' outcomes and reduce health expenditures. This requires reliable tools for early identification of at-risk patients and personalized interventions to tailor in-hospital care. Early identification of patients at high risk for transitioning to AKD (and eventually to CKD) is even more important to avoid the long-term sequelae of AKI.

Accumulating evidence supports the use of electronic health records (EHR) and the implementation of artificial intelligence (AI) to improve clinical decision-making [3,12–14]. In nephrology, machine learning (ML) holds the potential for early detection of kidney disease and as clinical decision support to optimize and personalize patient care [3,12,15–17]. Several studies evaluated ML approaches to predict AKI or AKD in specific clinical situations, particularly in the ICU [3,16,17]. Still, only a

few reports are available on the potential of ML to identify and predict the AKI to AKD transition in a broader spectrum of subjects with various medical conditions. Additionally, most studies rely on publicly available retrospective datasets, and results are cumbersome for implementation and use in real-world settings.

We aimed to expand current knowledge on using EHR and ML to predict AKI in an unselected cohort of hospitalized subjects (all hospitalizations in our hospital network) and to predict AKI-to-AKD transition among subjects experiencing AKI during the first 24 hours of hospital stay by longitudinally relaunching the ML model and integrating the laboratory data progressively accumulated during the first, second, third, and fourth days of hospitalization. This study, leveraging EHR, serves as a *proof of concept* for future integration of an ML approach into the patient's chart to timely identify AKI and accurately predict the risk of AKI-to-AKD transition. Abbreviations used in the text are listed in Table 1.

## Methods

This retrospective study included data from all consecutive inpatients to the Ente Ospedaliero Cantonale (EOC), a public hospital network serving approximately 350'000 people, between January 01 and December 31, 2022. The local Ethics Committee approved the study (date 02.06.2023, id 2023−00738). Data were accessed for research purposes on January 19, 2024. Demographic, clinical, and laboratory data were obtained from the hospital EHR and used to generate the study database (a brief description of the is provided in the S1 Supporting Information).All records were deidentified to protect patients' privacy. The Ethics Committee waived the patient's informed consent, considering the disproportionate effort to consent to all subjects of interest.

**Table 1. Abbreviations used in the text.**

| AKI | Acute kidney injury |
|---|---|
| ADQI | Acute Disease Quality Initiative |
| AI | Artificial intelligence |
| AKD | Acute kidney disease |
| AUC-ROC | Receiving Operator Curve |
| CKD | Chronic kidney disease |
| CKD-EPI | Chronic Kidney Disease Epidemiology Collaboration |
| eGFR | Estimated glomerular filtration rate |
| EHR | Electronic health records |
| EOC | Ente Ospedaliero Cantonale |
| ML | Machine Learning |
| NAKI | Normal renal function during the entire hospitalization |
| RRT | Renal replacement treatment |
| sCr | Serum creatinine |
| SHAP | Shapely additive explanations |
| AUC-PRC | Precision-recall curve |
| PPV | Positive predictive values |
| NPV | Negative predictive values |
| **Machine Learning models** | |
| AKI_HOSP_T0 | Launched at the patient's admission, predicts the occurrence of AKI during the entire hospital stay |
| AKD_T1 | Launched at day 1, predicts the AKI to AKD transition |
| AKD_T2 | Launched at day 2, predicts the AKI to AKD transition |
| AKD_T3 | Launched at day 3, predicts the AKI to AKD transition |
| AKD_T4 | Launched at day 4, predicts the AKI to AKD transition |

## Inclusion exclusion criteria

We considered all records of patients admitted to the four EOC hospitals in an acute care setting in 2022 (Fig 1). All subjects were treated according to standard care and followed from hospitalization to discharge. Records were excluded if the patient presented with CKD or severe renal function dysfunction before hospital admission, defined as an estimated glomerular filtration rate (eGFR) below 15 ml/min/1.73m$^2$(CKD stage 5), or received renal replacement treatment (RRT). Finally, records of repeated hospitalizations (i.e., re-hospitalizations within 18 days) were also excluded.

## Study aims

The aims of the study are as follows: **Aim 1**) to predict AKI during hospitalization based on pre-admission and admission data; **Aim 2**) to longitudinally predict AKI to AKD transition among patients experiencing AKI on the first day of hospitalization (*proof of concept*). We focused on subjects experiencing AKI during the first day to minimize right censoring due to patient mortality or discharge before the completion of 7 days of observation following the onset of AKI; **Aim 3**) to assess the feasibility of the ML approach to predict AKI and AKI to AKD transition at patient level (*proof of concept*).

## Definitions of AKI and AKD

Pre-hospitalization serum creatinine (sCr baseline) and all sCr values recorded during the hospitalization were used to define the renal function trajectory and the occurrence of AKI and AKD.

AKI was defined according to Kidney Disease Improving Global Outcomes (extended KDIGO) guidelines, [18], i.e., an increase in sCr by ≥26.5 µmol/L (0.3 mg/dl) within 48 hours or an increase in sCr to ≥1.5 times baseline sCr. Considering renal injury may occur before hospitalization and minimize the left censoring, a sCr decrease from baseline greater than 26.5 µmol/L (0.3 mg/dl) or a decrease greater than 1.5 times was also deemed evidence of AKI. AKI staging was

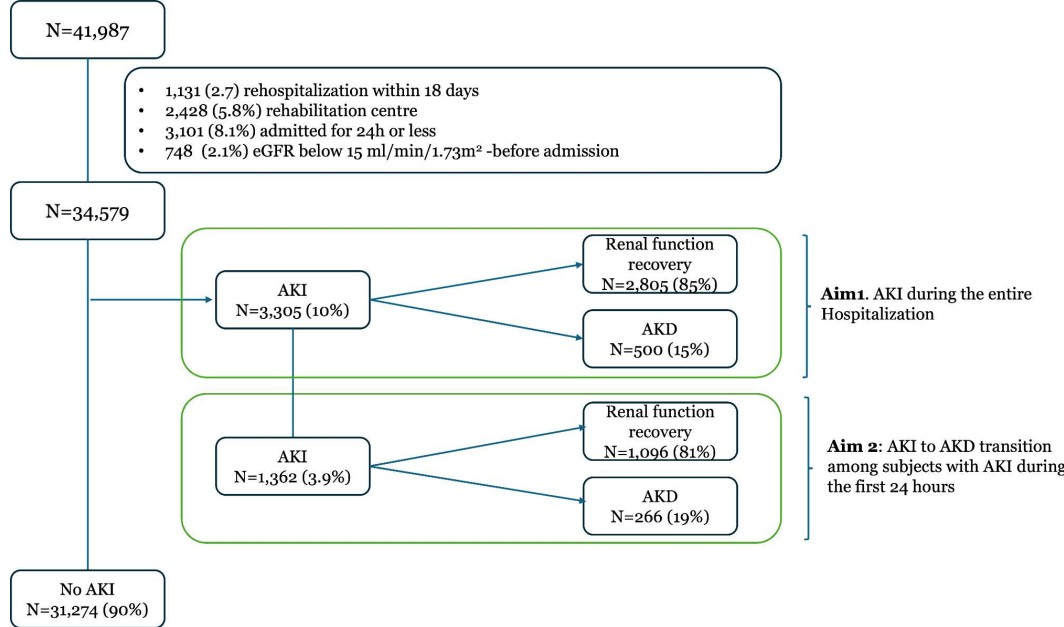

**Fig 1. The flow diagram describes the study cohort utilized for *Aim 1* and *Aim 2* according to renal function trajectories.** After the exclusion of records not fulfilling the inclusion and exclusion criteria, we utilized 34,579 records for *Aim 1* (described in the box with a light green outline) and 1,362 records for *Aim 2* (patients with AKI during the first 24 hours of hospitalization described in the light blue box).

performed according to the degree of sCr changes (S1 Supporting Information). Urine output was not used as a criterion to define AKI due to the lack of data on urine output at hospital admission in most patients. Of note, the use of extended KDIGO definition was previously shown to detect cases of AKI occurring before or earlier in the hospitalization, similarly associated with worse outcomes [19].

According to the 2017 Acute Disease Quality Initiative (ADQI) [1], AKD is defined as persistent renal damage and renal dysfunction (i.e., persistence of sCr > 1.5 times baseline value or sCr value greater than >353.6 μmol/L after seven days if baseline sCr not available) lasting 7–90 days after exposure to an AKI-initiating event [1]. (S1 Supporting Information).

Based on previous experience in which it was shown that the lowest value, rather than the time window before hospitalization, influences the yield of AKI [20], baseline sCr was defined as the most recent pre-admission sCr level measured within one year (365 days) and seven days before admission. In case no baseline sCr was retrieved, the diagnosis of AKI was based on the eKDIGO criteria as summarized in S1 Supporting Information.

Renal recovery was defined as a reduction in sCr within 7 days from AKI to a value less than 1.5 times the baseline value or a sCr value lower than 353.6 μmol/L if the baseline sCr is not available.

Based on renal function trajectories, three different patient groups are identified: i) patients with normal renal function during the entire hospitalization (NAKI); ii) patients with AKI and renal function recovery (AKI); iii) patients with AKI without renal function recovery within seven days from the establishment of the diagnosis of AKI (AKD) S1 Supporting Information.

## Statistical analysis and machine learning

We developed and retrospectively evaluated five ML models. Among these models, one is launched at patient admission to predict an AKI during the hospital stay (AKI_HOSP_T0). The other four models longitudinally evaluated the risk of AKI to AKD transition among subjects with AKI during the first day of hospitalization (AKD_T1, AKD_T2, AKD_T3, AKD_T4) (Fig 2).

All models utilized EHR data. AKI_HOSP_T0 used administrative data available at admission and pre-admission laboratory data dating back one year. Subsequent models (AKD_T1-4) also incorporated laboratory data accumulated during the hospital stay. Specifically, AKD_T1 included laboratory data until day one, AKD_T2 included laboratory data until day two, and so forth.

AKI_HOSP_T0 was based on the entire dataset of admitted patients (N = 34,579). Conversely, to evaluate the feasibility of a real-time data analysis approach (proof of concept) to improve AKD prediction AKD_T1-4 were based on patients who experienced an AKI onset on the first day of admission (N = 1362, N = 1263, N = 1157, N = 1043, respectively) (Fig 2). This approach was decided due to an overall mean hospitalization length shorter than 7 days required by the KDIGO definition of AKD. In case of death, patient discharge, or lack of renal function assessment seven days after AKI, AKD was defined according to the last sCr value available (right censoring). While this approach would enhance sensitivity, it would also increase the algorithm's negative predictive power (NPV), a desirable feature for a diagnostic tool designed to rule out patients with low or no risk of developing renal function impairment.

The data quality of the EHR was assessed through a multi-step approach, including pre-processing laboratory data (map, clean, and convert), matching and normalizing timestamps at the hospitalization level (t0 equals admission time), and organizing all the variables in a standardized format (i.e., each variable comes with a group, a description, and the normalized timestamp). No imputation on missing data was carried out.

The five models were obtained using the same ML algorithm and feature selection scheme and were validated through identical procedures. Specifically, we utilized LightGBM, a gradient-boosting framework employing tree-based learning algorithms. LightGBM is a popular choice for handling large datasets and is known for its efficiency in dealing with sparse data and effectively handling missing values. LightGBM was also used for feature selection. The process involves initially fitting the algorithm with all variables and retaining only the most essential variables. Subsequently, the algorithm is

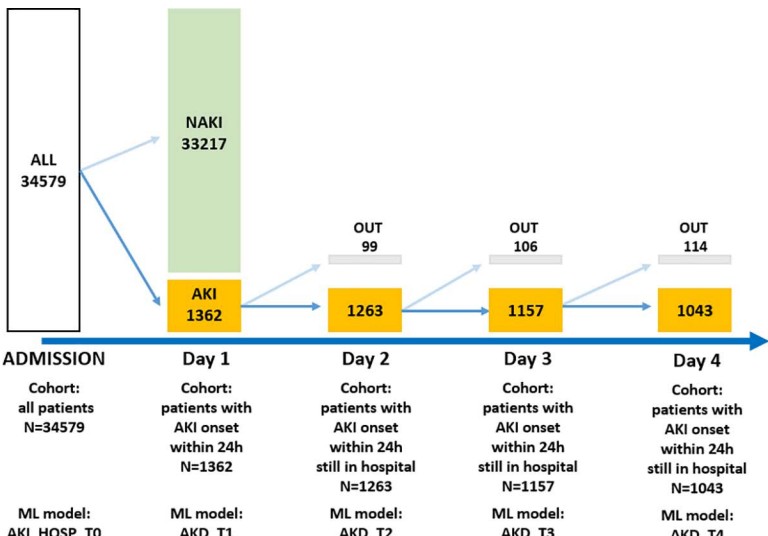

**Fig 2. illustrates the execution day and the number of patients used for each developed ML model.** Model AKI_HOSP_T0 predicts whether an admitted patient will develop an AKI during their stay. This model is executed at admission time and is developed using the entire dataset (N = 34,579). Models AKD_T1-4 predict whether patients who experience AKI within the first 24 hours of hospitalization will progress to AKD. AKD_T1 is based on patients still hospitalized on day 1 (24 hours – N = 1,362). AKD_T2 is based on patients still hospitalized on day 2 (48 hours – N = 1,263). AKD_T3 is based on patients still hospitalized on day 3 (72 hours – N = 1,157). AKD_T4 is based on patients still hospitalized on day 4 (96 hours – N = 1,053). Fig legend: AKI: acute kidney injury; NAKI: non-acute kidney injury; AKD: acute kidney disease; OUT: patients discharged or dead.

refitted using only the selected variables. This approach ensures the model focuses on the most relevant features identified by the LightGBM algorithm's intrinsic feature importance evaluation. Based on our experimental results, it was determined that retaining ten variables was a reasonable choice.

The dataset was divided into a training and validation set (90%) and an external validation set (10%). The chronological order of cases was maintained, with newer cases assigned to the external validation set to better mimic real-world conditions. The training and validation set was utilized to evaluate various hyperparameters of the LightGBM (LGB) model through k-fold cross-validation (3 repetitions), randomly generating hyperparameter configurations. The best-performing configuration, identified as having the highest mean AUC-PRC, was chosen as the best model. This model was trained on the entire training and validation set and calibrated to refine its predicted probabilities, ensuring they accurately reflect the actual likelihood of outcomes. The calibrated model, referred to as the ML model, was then assessed for its performance on the external validation set. (S1 Supporting Information) provides a graphical representation of ML pipeline. The evaluation process included calculating four key performance metrics: AUC-ROC, area under the receiver operating characteristic curve (AUC-PRC), positive predictive value (PPV), and negative predictive value (NPV). PPV and NPV were computed using a threshold determined by the g-mean method, which effectively balances sensitivity and specificity (Supplemental 2 Fig) [21]. AUC-ROC and AUC-PRC provide valuable insights into the model's ability to discriminate between classes and their performance across different thresholds. A higher AUC-ROC indicates better overall classification performance, while a higher AUC-PRC signifies better precision-recall trade-off, particularly important in imbalanced datasets. PPV and NPV provide a means to assess the algorithm's performance using a predetermined threshold.

Shapely additive explanations (SHAP) have been used to explain the models trained on the complete dataset. The contribution of each feature and the direction (positive or negative) of its effect have been calculated.

## Results

A total of 41,987 consecutive hospitalization records from 2022 were analyzed. Records were excluded if the following criteria were met: re-hospitalizations within 18 days (1,131–2.7%), admission of subacute or rehabilitation care (2,428 - 5.8%), hospitalizations shorter than 24 hours (3,101–8.1%), and eGFR below 15 ml/min/1.73 m$^2$ before hospital admission (748 - 2.1%). Thus, 34,597 (82.3%) records were available for further analysis (Fig 1). The mean age was 60.8±25.3, and 50% were females; the patient's characteristics at hospital admission are summarized in Table 2. Baseline sCr value was available in 48% (N=16,597) of the cases. The case-mix of subjects with AKI during the first 24 hours is summarized in S1 Supporting Information.

### AKI and AKD occurrence and association with outcomes

About 10% (N=3,305) of subjects developed AKI during hospital stay. Most AKI episodes were recorded during the first three days of hospitalization: 1,362 (41%) occurred Within the first 24 hours, 2,279 (69%) within 48 hours, and 2,556 (77%) within 72 hours after admission (S1 Supporting Information).

Overall, of the 3,305 subjects experiencing AKI, 15% (N=500) transitioned to AKD (Fig 1). Similarly, of the 1,362 subjects with AKI during the first 24 hours utilized for the analysis of the aim 2, 266 transitioned to AKD (Fig 1). Not all AKD subjects were followed for at least seven days after the occurrence of AKI and of the 266 cases of AKD considered, 139 (52.2%) were discharged before the completion of the 7th day from AKI. The median follow-up time was 3.9 days from AKI (S1 Supporting Information).

Table 2. Demographic, administrative, and laboratory data of the entire study cohort (utilized for *Aim 1*).

| Variable | ALL | NAKI | AKI | AKD |
|---|---|---|---|---|
| | (N=34,579) | (N=31,274) | (N=3,305) | (N=500) |
| Age, Mean±SD | 60.82±25.37 | 59.40±25.85 | 74.17±14.62 | 74.32±15.16 |
| Missing | 0 (0%) | 0 (0%) | 0 (0%) | 0 (0%) |
| Female sex, % | 50% | 51% | 40% | 40% |
| Missing | 0 (0%) | 0 (0%) | 0 (0%) | 0 (0%) |
| Medical Speciality Medicine, % | 31% | 29% | 53% | 59% |
| Missing | 0 (0%) | 0 (0%) | 0 (0%) | 0 (0%) |
| Medical Speciality Urology, % | 6% | 6% | 9% | 5% |
| Missing | 0 (0%) | 0 (0%) | 0 (0%) | 0 (0%) |
| Urgency admission, % | 62% | 59% | 85% | 84% |
| Missing | 0 (0%) | 0 (0%) | 0 (0%) | 0 (0%) |
| Elective admission, % | 33% | 35% | 14% | 14% |
| Missing | 0 (0%) | 0 (0%) | 0 (0%) | 0 (0%) |
| Institute Emergency Medicine, % | 6% | 5% | 16% | 11% |
| Missing | 0 (0%) | 0 (0%) | 0 (0%) | 0 (0%) |
| Institute Surgery, % | 35% | 36% | 26% | 20% |
| Missing | 0 (0%) | 0 (0%) | 0 (0%) | 0 (0%) |
| sCr baseline, Mean±SD | 81.00 [66.00–101.00] | 80.00 [66.00–98.00] | 93.00 [72.00–125.00] | 82.00 [64.00–108.00] |
| Missing | 17938 (52%) | 16789 (54%) | 1149 (35%) | 41 (8%) |
| sCr 1st, Mean±SD | 84.00 [68.00–107.00] | 81.00 [67.00–99.00] | 130.00 [98.00–180.00] | 130.00 [95.00–223.00] |
| Missing | 10155 (29%) | 10155 (32%) | 0 (0%) | 0 (0%) |

Table legend: data are presented as mean(standard deviation) or median[interquartile range] when appropriate. sCR: serum creatinine at baseline; sCr 1st: first serum creatinine available during hospitalization.

When sCr data were cross-checked with administrative data, only 1,166 out of 3,305 (35%) and 261 out of 500 (52%) cases of AKI and AKD were categorized and coded among the diagnoses at patient discharge. Only 4% and 8% of hospitalizations complicated with AKI and AKD were referred to the nephrology unit (data available only for the main hospital in Lugano).

As expected, we observed a graded association between AKI severity and the transition to AKD. Three hundred fifty-four out of 2,954 (12%), 97 out of 245 (40%), and 49 out of 106 (46%) individuals with AKI stages 1, 2, and 3 transitioned to AKD, respectively.

Regardless of etiology and AKI severity, renal function impairment during hospitalization was associated with unfavorable outcomes and prolonged hospitalizations. The mean duration of the hospitalization was 5.9±6.7, 13.1±12.5, and 13.9±15.1 among individuals without renal dysfunction (NAKI), AKI, and AKD subjects, respectively (S1 Supporting Information). Similarly, the unadjusted overall mortality was 2%, 7%, and 32% among NAKI, AKI, and AKD, respectively.

### Aim 1: ML model to predict the occurrence of AKI during hospitalization

Based on demographic, laboratory, and administrative data from one year before and at the time of hospital admission, the ML model to predict AKI during the entire hospitalization (AKI_HOSP_T0) achieved a high AUC-ROC (0.80) and NPV (0.96); however, these values are likely inflated due to the highly imbalanced dataset (10% AKI prevalence). While the PPV (0.22) is low, it remains reasonable given the class imbalance. Sensitivity (0.72) and specificity (0.70) are moderate (Table 3 and Fig 3). Notably, these results were obtained using only data available at the time of admission and apply to the entire acute hospital population, excluding only patients with chronic renal dysfunction.

The SHAP analysis reveals that the most relevant features associated with the risk of AKI are age, baseline renal function, the maximum value of creatinine within one year of hospital admission, elective versus urgent hospital admission, and the type of admission ward (Fig 4). Similar results were observed when the analyses were restricted to subjects with AKI during the first 24 hours of hospitalization (S1 Supporting Information).

### AIM 2: ML models to predict an AKI to AKD transition among subjects with AKI during the first 24 hours of hospitalization

Of the initial 1,362 subjects experiencing AKI during the first 24 hours of hospital stay, 99 (7.3%), 106 (7.8%), and 114 (8.4%) either expired or were lost due to hospital discharge at day 2, 3, and 4, respectively (Fig 2).

**Table 3. Diagnostic accuracy calculated on the external validation set of the ML model predicting the risk of developing an AKI during the hospitalization based on data available before and at hospital admission (model: AKI_HOSP_T0). AKI: acute Kidney Injury, ROC: Receiving Operator Curve, PRC: Precision-Recall Curve, AUC: Area Under the Curve, PPV: Positive Predictive Value, NPV: Negative Predictive Value.**

|  | AKI during hospitalization |
|---|---|
| Model | AKI_HOSP_T0 |
| Hospitalizations | 34,579 |
| AKI Cases | 3,305 |
| AKI prevalence | 10% |
| **Metrics** |  |
| AUC-ROC | 0.80 |
| AUC-PRC | 0.30 |
| PPV | 0.22 |
| NPV | 0.96 |
| Sensitivity | 0.72 |
| Specificity | 0.70 |

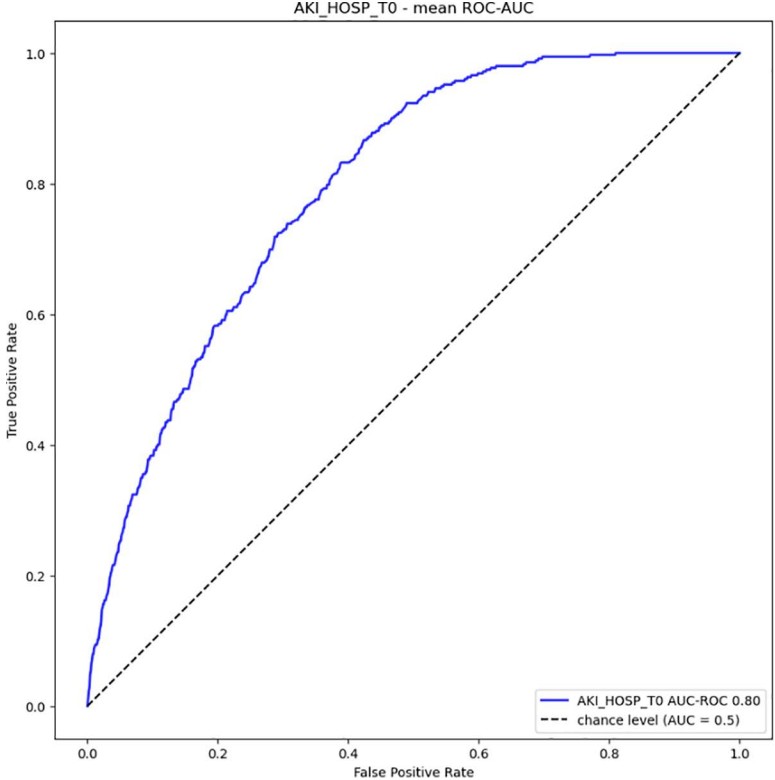

**Fig 3. Receiving Operator Curve (ROC) of the ML model predicting an AKI during the entire hospitalization (AKI_HOSP_T0).** The predictions are made at the time of patient admission.

Based on data demographic, laboratory, and administrative data from one year before and at the time of hospital admission, along with laboratory data progressively accumulated during the hospital stay, the ML models to predict the transition to AKD (AKD_T1 to AKD_T4) presented a high AUC-ROC (0.78–0.89) and NPV (0.93–0.99); however, these values are likely inflated due to the highly imbalanced dataset (17−20% AKI prevalence). While the PPV (0.40–0.47) is low, it remains reasonable given the class imbalance. Sensitivity (0.79–0.94) ranges from moderate to very good. Specificity (0.70–0.78) ranges from moderate to solid. Importantly, the results enhance as more data is introduced, with the model achieving its best performance on day 4 (Table 4 and Fig 5).

The SHAP analysis reveals that the most relevant features associated with the risk of AKD are the changes (delta change, speed of change) in renal function and associated laboratory abnormalities (electrolytes, inflammatory index, red and white blood cell abnormalities) (S1 Supporting Information).

### Aim 3: Patient level analysis to predict AKI and AKI to AKD transition

The proposed ML models can be applied at the patient level, offering risk predictions and contributing factors explanations. Fig 6 represents the output produced by the 5 models for a patient developing AKI during the first 24 hours of hospitalization (sCr from 100 to 135 mmol/L) and AKD (sCr > 1.5-fold from baseline after seven days). Upon patient admission, the AKI_HOSP_T0 model accurately predicted the likelihood of patients developing acute kidney injury (AKI) during their hospital stay. The most significant factors influencing this prediction were the patient's age (92 years), baseline renal function, maximum serum creatinine value in the year before hospitalization, and emergency hospitalization (Fig 7A).

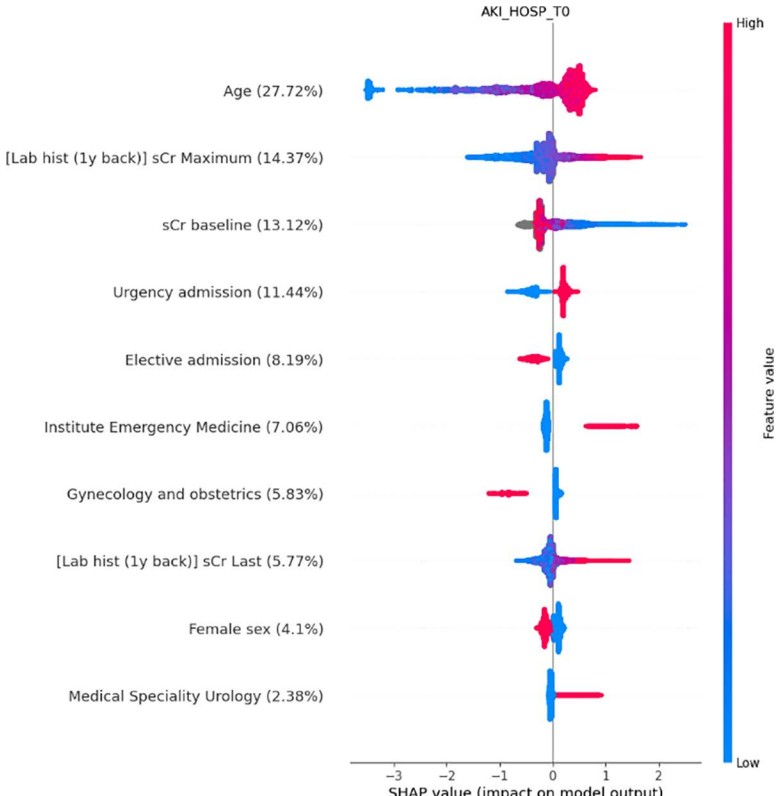

**Fig 4. SHAP analyses of the ML model running at admission and predicting an AKI during the entire hospitalization (AKI_HOSP_T0).** Variables are listed in descending order of importance, with the most important variable at the top. The relative importance of each variable is indicated in parentheses. Fig legend: sCr: serum creatinine; [lab hist (1y back)] sCr: maximum value of serum creatinine in the time window of 365 to 7 days prior to hospitalization; [lab hist (1y back)] sCr Last: last value of serum creatinine in the time window of 365 to 7 days prior to hospitalization.

Models AKD_T1, AKD_T2, AKD_T3, and AKD_T4, launched from day one and in the following three days post-onset of AKI, suggested a high risk of renal function impairment prolongation over the seventh day of AKI (Fig 7B–7E). These last predictions were mainly based on the sCr trajectories documented during the first four days of observation (135 mmol/L, 196 mmol/L, and 214 mmol/L) (Fig 7B–7E).

## Discussion

This retrospective observational study utilized EHR accumulated the year before and at hospital admission to identify AKI during hospitalization. Unlike other studies [3], baseline renal function was available in most cases (>50%). Also, we longitudinally followed individuals with AKI during the first 24 hours to detect the AKD transition during the hospital stay. This study design was decided in consideration of the high prevalence of AKI episodes during the first days of hospitalization and the relatively short in-hospital follow-up of patients with AKI (i.e., less than seven days from the AKI episode).

Although significant variability in the incidence of AKI is reported depending on the settings and case-mix, our results corroborate what was reported by others [3,4,10], showing that 10% of all hospitalized subjects were diagnosed with AKI, and 15% of these developed AKD during the hospital stay. We confirmed that AKI and AKD are associated with prolonged hospitalizations and a substantial and graded increased risk of mortality, regardless of the severity, duration, and etiology of renal dysfunction. While the study design may lead to an overestimation of AKD cases, it also indicates that renal function is not always followed until recovery before patients' discharge. Consistently, we also observed an undercoding of AKI

**Table 4. Diagnostic accuracy of the ML models in predicting AKI to AKD transition among subjects experiencing AKI during the first day of hospitalization calculated on the external validation set. AKI: acute Kidney Injury, ROC: Receiving Operator Curve, PRC: Precision-Recall Curve, AUC: Area Under the Curve, PPV: Positive Predictive Value, NPV: Negative Predictive Value.**

| Model | Execution time (day) | N | Metric | |
|---|---|---|---|---|
| AKD_T1 | 1 | 1,362 | AUC-ROC | 0.78 |
| | | (prevalence 20%) | AUC-PRC | 0.47 |
| | | | PPV | 0.40 |
| | | | NPV | 0.93 |
| | | | Sensitivity | 0.79 |
| | | | Specificity | 0.70 |
| AKD_T2 | 2 | 1,263 | AUC-ROC | 0.79 |
| | | (prevalence 18%) | AUC-PRC | 0.42 |
| | | | PPV | 0.36 |
| | | | NPV | 0.96 |
| | | | Sensitivity | 0.87 |
| | | | Specificity | 0.63 |
| AKD_T3 | 3 | 1,157 | AUC-ROC | 0.83 |
| | | (prevalence 17%) | AUC-PRC | 0.5 |
| | | | PPV | 0.41 |
| | | | NPV | 0.96 |
| | | | Sensitivity | 0.86 |
| | | | Specificity | 0.71 |
| AKD_T4 | 4 | 1,043 | AUC-ROC | 0.89 |
| | | (prevalence 17%) | AUC-PRC | 0.61 |
| | | | PPV | 0.47 |
| | | | NPV | 0.99 |
| | | | Sensitivity | 0.94 |
| | | | Specificity | 0.78 |

diagnoses and lower-than-expected referrals to the nephrologist for these cases, corroborating the notion of AKI/AKD as an unmet medical need. However, others have reported a discrepancy in the prevalence of renal dysfunction depending on the source of data (administrative vs. laboratory data) used to define it AKI [11].

Our findings expand the available evidence on using ML for early AKI diagnosis [3,12,22,23]. The multiple etiologies of AKI/AKD complicate risk prediction and classification, and the dynamic changes in clinical scenarios require continuous monitoring to predict renal prognosis. Traditional prediction models are usually static and based on average risk estimation in specific groups of individuals [16]. In contrast, AI, encompassing dynamically various data sources, yields mild to modest discriminative power in AKI detection (ROC from 0.65 to 0.80) [16], and ML models allow for risk prediction at the individual level, suggesting promise in AKI and AKD prediction [16]. By relaunching the ML model every 24 hours and adding more laboratory data to the models as they become available, our approach allows for predicting AKI to AKD transition about three to seven days before the occurrence with a good and progressively improved discriminative power (from 0.78 to 0.89).

By mimicking the doctor's practice, ML allows for accurate risk prediction also at the individual level. Notably, the model's negative predictive power at each time point suggests that this approach is valuable in ruling out individuals at low risk for AKI and, if this occurs, individuals at low risk for AKI to AKD transition. This feature is particularly significant given the relatively low incidence of these conditions within the overall study cohort. The longitudinal integration of our ML models

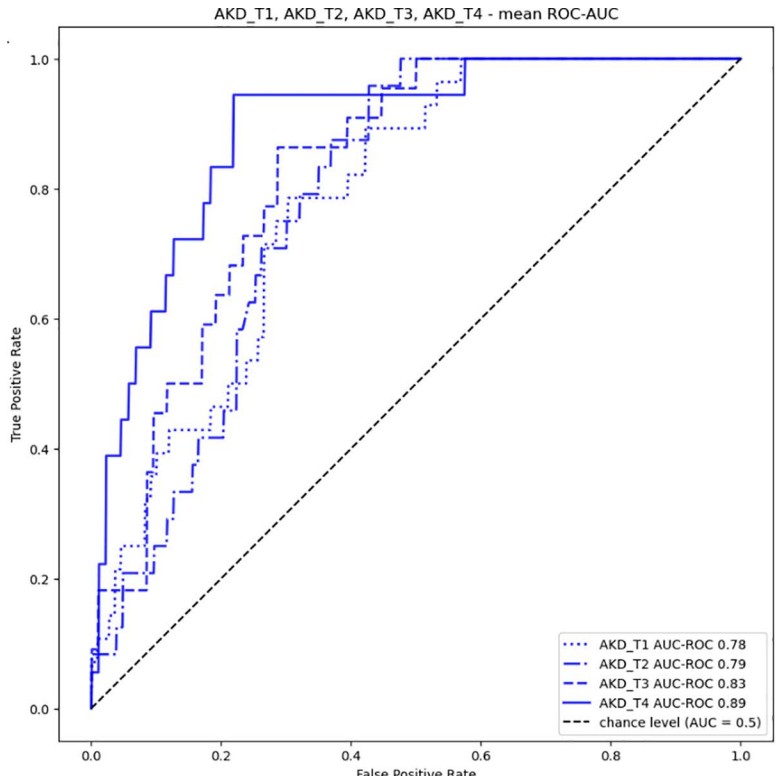

**Fig 5. Receiving Operator Curve (ROC) of the ML models to predict the risk of AKD transition at day 1, 2, 3, and 4 (AKD_T1 through T4).**

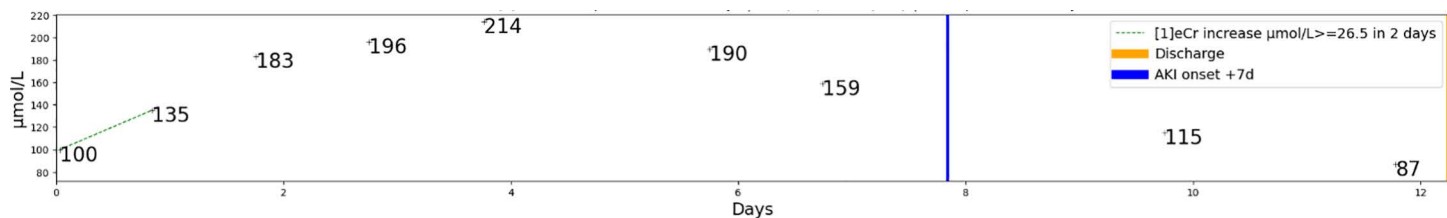

**Fig 6. Represents a single case of a patient developing AKI during the first 24 hours of a hospital stay.**

could function as a screening tool to identify low-risk patients and prioritize high-risk cases for early evaluation, guiding decisions regarding serum creatinine tests or additional renal function assessments. This is especially valuable in light of the diagnostic delay associated with the rise in serum creatinine that occurs 48–72 hours after renal injury. By excluding low-risk individuals for AKI, attention can be focused and renal function monitoring intensified on the few individuals (about 1 in 10) who will develop clinically manifest renal damage, with the potential of transitioning to AKD (about 1 in 100).

The potential benefits of the ML approach go beyond patient identification and may promote individualized and risk-based patient care. Indeed, the dynamic integration of demographic, administrative, and laboratory data to predict real-time individual renal function trajectories provides physicians with hints (and potential therapeutic targets) on what factors may be more closely related to AKI and AKD. Future applications are needed to test whether this approach promotes individualized care, optimizes resource allocation, and reduces the length of hospitalization and renal failure-associated costs.

 

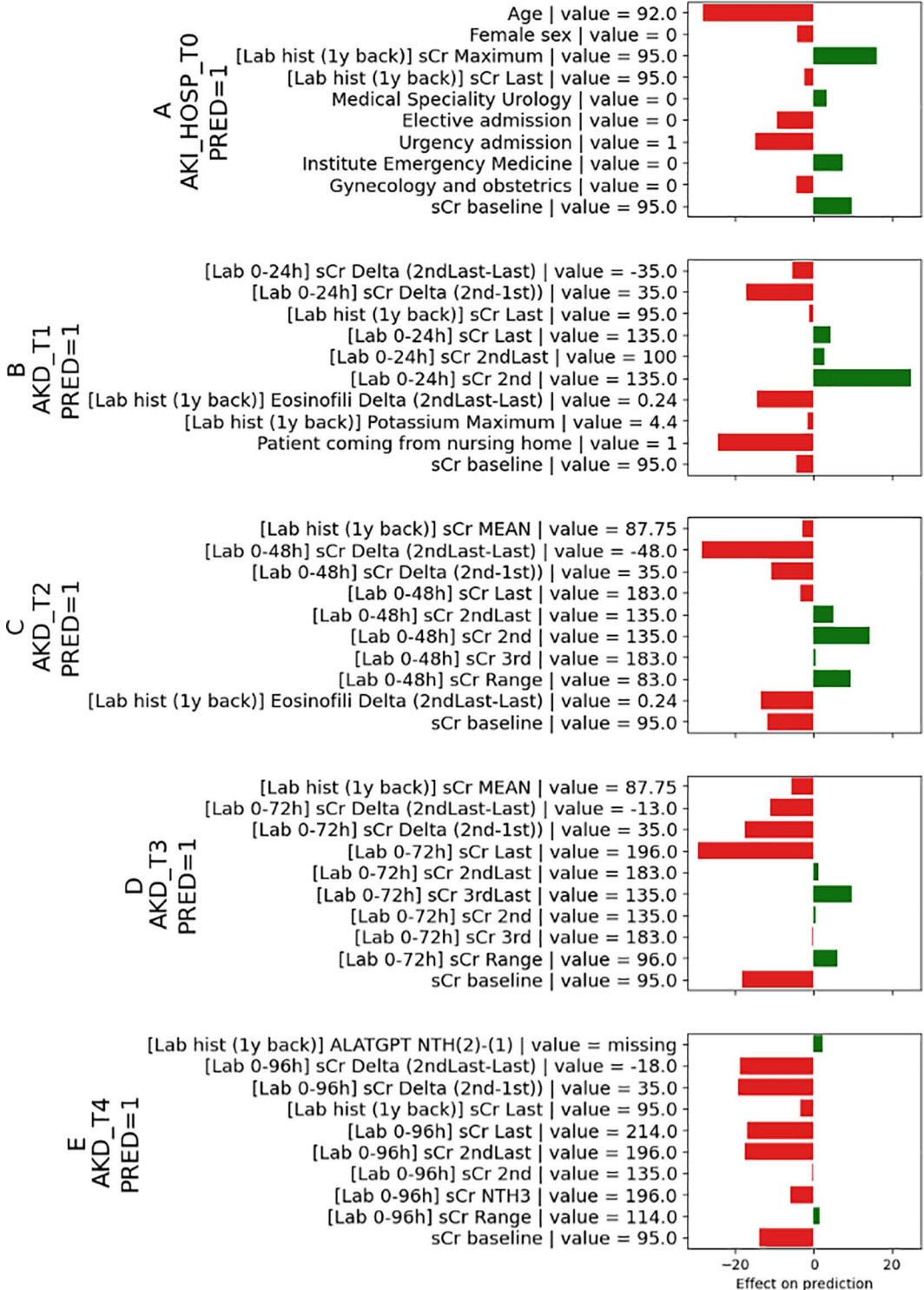

**Fig 7. represents the patient-level SHAP analyses of the ML model running at admission and predicting AKI during the hospitalization** (A) and the risk of AKD transition at day 1, 2, 3, and 4 (B-E). Model A predicted AKI (PRED = 1). Model B predicted no transition to AKD (PRED = 0). Models C-E predicted the transition to AKD (PRED = 1). Color legend: In red are the factors positively associated with the risk of the outcome of interest; in green are the factors negatively associated with the outcome of interest. Fig legend: sCr: serum creatinine; ALATGPT NTH: alanine aminotransferase; STD: Standard deviation. Between the square brackets, the time point considered is indicated. For example, [lab hist (1y back)]sCr maximum refers to the

maximum serum creatinine value in the 365 to 7 days before hospitalization. Delta signifies a change that occurs during hospitalization. The time points used to calculate the delta are indicated in round brackets. For example, [lab 0–24] sCr Delta (2nd-1st) refers to the change in serum creatinine (micromol/l) between day 2 and day 1.

Our study has some limitations. First, this retrospective *proof-of-concept* study utilizes data from a single healthcare system, and information on the etiology of renal failure is missing. While this last piece of information would likely have enhanced the model's accuracy, our sequential approach, utilizing real-world data, demonstrates its potential in predicting the transition from AKI to AKD overcoming this inherent data limitation. Second, a proper external validation was not carried out. However, we adopted a standard approach of dividing the dataset into a training and a validation set. Although both the training and validation sets are subsets of the original dataset, they are independent. This methodology offers insights into the model's robustness and reliability potential and represents the base for future, prospective and appropriately designed studies. Third, data on comorbid conditions or concomitant medications were not considered. While including more variables may improve model accuracy, it may also result in model overfitting, diminishing its generalization and application in diverse contexts. Fourth, various ML models are available, and more accurate approaches may exist to predict AKI to AKD. However, a higher degree of complexity also reduces the exploitability of the AI model. Fifth, we should have investigated the impact of gender and/or different treatments on renal function. While this information would add to the current endeavor, the relatively small data available would likely limit our model accuracy, repeatability, and exportability. Also, these analyses were outside the scope of this study. Sixth, caution should be exercised when evaluating the results of this study. The absence of proper external validation and the exclusion of subjects with pre-existing advanced renal function impairment likely biases our study cohort towards a low-risk population for AKI or AKD, potentially increasing the NPV and limiting the algorithm's generalizability. While the use of this approach to individuals with advanced CKD or significant comorbidities cannot be extrapolated, our approach underscores an unmet medical need, and it offers the opportunity to design future prospective studies with protocolized interventions for subjects at high risk of major renal events. However, despite these limitations, this study is one of the first attempts to utilize ML to predict AKI to AKD transition in hospitalized patients in a longitudinal fashion.

In conclusion, leveraging a simple and replicable ML model approach on EHR data allows for accurate AKI and AKD prediction in hospitalized subjects. Event prediction through AI may help individualize patient care and reduce the cost burden associated with AKI and AKD. However, without appropriate external validation, caution must be exercised in interpreting the current results. Future efforts and specific initiatives are required to ascertain whether the implementation of an AI decision-support tool can enhance patient care, improve outcomes, and optimize healthcare resource allocation.

## Supporting information

**S1 File. Supporting information.**
(DOCX)

## Author contributions

**Conceptualization:** Lorenzo Ruinelli, Antonio Bellasi.

**Data curation:** Lorenzo Ruinelli.

**Formal analysis:** Lorenzo Ruinelli.

**Methodology:** Antonio Bellasi.

**Supervision:** Clelia Di Serio, Antonio Bellasi.

**Writing – original draft:** Lorenzo Ruinelli, Chantal Sieber, Antonio Bellasi.

**Writing – review & editing:** Pietro Cippà, Chantal Sieber, Clelia Di Serio, Paolo Ferrari.

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
