## [Decision Letter · Decision Letter 0]

Dear Dr. Bellasi,

Thank you for submitting your manuscript to PLOS ONE. After careful consideration, we feel that it has merit but does not fully meet PLOS ONE’s publication criteria as it currently stands. Therefore, we invite you to submit a revised version of the manuscript that addresses the points raised during the review process.

We look forward to receiving your revised manuscript.

Kind regards,

Fabio Sallustio, PhD

Academic Editor

PLOS ONE

Journal Requirements:

Additional Editor Comments:

Please, address the methodological issues of the model raised by reviewers, including the calibration and the external validation.

Reviewers' comments:

Reviewer's Responses to Questions

**Comments to the Author**

1. Is the manuscript technically sound, and do the data support the conclusions?

Reviewer #1: Yes

Reviewer #2: Yes

Reviewer #3: No

Reviewer #4: Yes

2. Has the statistical analysis been performed appropriately and rigorously?

Reviewer #1: Yes

Reviewer #2: Yes

Reviewer #3: No

Reviewer #4: Yes

3. Have the authors made all data underlying the findings in their manuscript fully available?

Reviewer #1: Yes

Reviewer #2: Yes

Reviewer #3: Yes

Reviewer #4: Yes

4. Is the manuscript presented in an intelligible fashion and written in standard English?

Reviewer #1: Yes

Reviewer #2: Yes

Reviewer #3: Yes

Reviewer #4: Yes

Reviewer #1: Hello! I just reviewed the article entitled "Usability of Machine Learning Algorithms based on Electronic Health Records for the Prediction of Acute Kidney Injury and Transition to Acute Kidney Disease: a Proof of Concept Study" by Lorenzo Ruinelli

et al. With this study, the authors proved that ML algorithms could predict AKI and/or AI to AKD transition. Overall, the study is well-written, in good English, with a goo flow and logical. Before considering publishing the article, I have some suggestions for the authors, as minor revisions:

1. Please revise the continuous variables regarding the normal distribution. I saw some high SD compared to the means and this suggest that the data is not normally distributed.

2. In table 2 and supplemental table 2, you present the estimated GFR 1st. I assume this is the calculated GFR from day 1 of hospital admission. Due to the fact that most of the patients presented already with AKI in the first day of hospital admission, I consider the use of eGFR to be inappropriate. The use of eGFR in context of acute kidney injury is unreliable. Nevertheless, you could replace the eGFR from day one with baseline eGFR - calculated with from the baseline creatinine. In this case, I would expect that patients who develop AKD to present lower baseline eGFR.

3. An important factor for transition to AKD from AKI would be the AKI cause. I consider that you should add in the limitations paragraph the absence of AKI cause in your model. Intrinsic AKI is proved to increase the risk of AKI prolongation towards AKD. Given this aspect, you indeed presented in the discussion section the clinical importance of AKI and/or AKI to AKD progression. I consider you could expand this paragraph and discuss the clinical solutions stratified by AKI causes (prerenal, intrinsic and postrenal - as many of AKI were from the urology department). Nevertheless, you should discuss the importance of the nephrotoxic medication avoidance in context of AKI.

4. Overall, a good article, logical and great flow. In the end, I would like to add that besides the diagnosis and prediction of AKI/AKD occurrence, the recognition of AKI is of great importance. As you clearly stated in your article, the AKI diagnosis rate was reduced and the nephrology reference was even lower.

5. Good luck!

Reviewer #2: Dear Authors,

I have read your work with great interest. It presents a highly intriguing application of AI in a very active research area. However, I would like to offer some observations and requests to enhance the paper’s clarity, which might be challenging to follow, and to highlight aspects that I believe could be valuable for the readers:

Incidence of AKI

The AKI incidence reported in your study appears to be lower than what is typically documented in the literature, which estimates it to be around 20% in hospitalized patients (DOI: 10.1016/S0140-6736(16)30240-9). Do you have any opinions or comments on this discrepancy?

Definition of AKI

The AKI definition used in your study is unclear. According to KDIGO guidelines, AKI diagnosis requires a creatinine increase within 48 hours or over 7 days from a baseline value. In your study, however, the baseline value could date back up to one year. Additionally, when no baseline value was available, you diagnosed AKI based on a two-day creatinine increase, but what was this increase compared against?

Definition of Renal Recovery

How was renal recovery defined in your study?

AKI Analysis

It is important to emphasize that AKI occurring during the first days of hospitalization differs from that developing later. Notably, there is a well-documented distinction between community-acquired AKI and hospital-acquired AKI, which may have different causes and outcomes (DOI: 10.1053/j.ajkd.2023.10.009, DOI: 10.2215/CJN.07920713).

For example, a recent prospective study demonstrated that the primary risk factor for community-acquired AKI is the use of nephrotoxic drugs (DOI: 10.1016/j.ejim.2024.09.016). However, this study lacks a detailed analysis of AKI causes and does not consider this factor.

Another observation is related to the models used to assess AKD risk in this study. These models are based on data collected during the early days of hospitalization, excluding AKI cases that develop later—cases that represent true hospital-acquired AKI and are generally associated with worse prognoses.

I suggest that the authors discuss these aspects in more detail (some of which are already partially addressed) and include them as study limitations.

Comparison of AKI and AKD Patients

A comparison between AKI patients who recover and those who progress to AKD would be interesting. Currently, Supplementary Table 2 lacks statistical analysis for this comparison.

SHAP Analysis Variables

The SHAP analyses include variables that are not explained, such as sCr MAX/SD/LAST. What do these variables represent, and how do they differ from sCr baseline? Additionally, how are the deltas calculated? Providing explanations for these factors would enhance the paper’s comprehensiveness.

Discrepancy Between Administrative and Biochemical Data

A recent study highlighted the discrepancy between administrative and biochemical data in AKI diagnosis and showed the existence of different patterns of AKI recognition in hospitals (DOI: 10.1093/ckj/sfae231). Including a discussion on this point could be useful, particularly for guiding future research.

Supplementary Figure 1 is unclear and could benefit from further clarification.

Reviewer #3: The study concerns a topic not treated in deep, which is of interest in a large number of patients admitted out of Intensive Care Units.

I have some key questions on the methods and results of this study.

The process of evaluation of machine learning algorithms is lacking of two steps: the first is the calibration and the second is the external validation.

The calibration test which is useful to calibrate the predictive model.

The external test set, which is necessary to prove the model performance and its ability to generalize on different geographical populations.

Consequently I suggest to review in depth the process of ML algorithms generation, because we don’t know if the model is well calibrated and if the algorithm can be widely applied.

Other aspects:

1) AKI and AKD occurrence and association with outcome. The authors report 3305 subjects who developed AKI during hospital stay. Then they report AKI in the first 24h = 1362 pts, in the 48 h 2279 and in the 72 hrs 2556. I think that it should be better to describe AKI periods: 0-24hrs, 25-48 hrs, 49-72 hrs, > 72hrs. This information can be added to supplemental figure 2.

2) A better explanation is needed for AKD transition. In this case it is not clear the definition of AKD because only 47.8% of patients with AKI in the first 24 hrs completed 7 days of observation needed for AKD assessment (see definition of AKD reported on methods sections).

3) Positive predictive value is reported, but the percentage is low. A comment is needed. Could we consider useful a prediction model with such as margin of error? In discussion this aspect is not reported or commented.

Reviewer #4: The study is very articulate and analyzed an interesting role for ML in predicting AKI or AKD occurrence in acute hospitalized patients. Nevertheless, some methodological limits (e.g. retrospective, mono-centric and selection criteria) I think this study may add interesting insights for ML in clinical practice. Below, the Authors can find my comments about the manuscript.

Minor revisions

A) Abstract

- "AKI and AKD" are used appropriately, but the definition of these conditions could be included, considering that the abstract is addressed to an audience that may not be familiar with these conditions. (e.g. According to the KDIGO guidelines?). Is it due to the word-limit?

- I suggest reformulating as follow: “…The majority of AKI episodes (77%) occurred within the first three days of hospitalization”.

- “AKI and AKD complicated 10% and 1.5% of hospitalizations”: respectively?

B) Aims

-Better “The aims of the study are as follows:” and “to minimize right censoring due to patient mortality or discharge before the completion of 7 days of observation following the onset of AKI;”.

Major revisions

Some comments for the Authors, to which I would appreciate their responses.

1) The main limitation of the Study may be the reduced representativeness of the enrolled patient population, as it excludes groups with significant renal comorbidities. However, if the aim is to develop a highly specific model for predicting AKI in a low-risk population, this exclusion may be appropriate." Indeed, by excluding patients with severe pre-existing renal disease, the dataset likely includes a higher proportion of individuals with low-risk for AKI or AKD and potentially increasing the NPV. This may limit the ML model's generalizability, as excluding patients with severe renal impairment reduces dataset representativeness for populations with higher prevalence of CKD or significant comorbidities

2) The exclusive use of sCr to define AKI and AKD, while widely accepted, may not fully include the complexity of AKI. sCr is influenced by factors such as age, sex, muscle mass, and hydration status, potentially distorting renal function assessment. Additionally, in patients with pre-existing conditions like dehydration or malnutrition, sCr may not accurately reflect renal function. For this reason, it would be interesting to have at least the reason for hospitalization and some information on comorbidities to better guide the clinical utility of the ML model.

3) The extended definition of AKI, including serum creatinine increases >26.5 μmol/L or >1.5 times baseline, may lead to overdiagnosis by identifying minor, clinically non-significant renal function fluctuations as AKI. While it improves early detection, these minor changes often may lack prognostic relevance, potentially increasing sensitivity at the expense of specificity affecting the clinical power of ML. Please, Authors comment this aspect in the text.

4) The retrospective design depends on potentially incomplete EHR data, which may affect accuracy. This could be a limitation for generalization that needs to be better considered in the discussion.

5) Right censoring due to death, discharge, or missing renal data within 7 days could skew results. The progressively smaller datasets for AKD models (T1-T4) may reduce ML model’s stability and generalizability. Additionally, time-specific models may fail to capture slower or atypical disease progressions, limiting their applicability.

In details, other comments about Discussion and Conclusion that I underline to the Authors.

- Only 35% of AKI and 52% of AKD cases were recorded, with a notably low rate of nephrology referrals. Could this suggest a significant proportion of events clinically not impactful?

- The median follow-up duration of 4.1 days, combined with right censoring due to discharge or death, may have led to an overestimation of the ML’s ability to predict long-term AKD outcomes. This requires further discussion.

- Serum creatinine (sCr) data were available in only 48% of cases. This limitation could significantly impact the ML model’s accuracy and should be addressed in the discussion.

- The AUC-ROC values (0.76–0.88), alongside missing data on patient comorbidities and medication use, are critical limitations that could affect both the ML model's performance and its clinical applicability. These aspects deserve more comprehensive consideration.

- The study is based on retrospective data from a single healthcare system. This raises concerns about the generalizability of the findings, which should be better contextualized in the discussion.

- Finally, it may be valuable to explore the relationship between hospitalization length and the occurrence of AKI in more detail. That Said, could Supplemental Figure 3 be integrated into the main text to provide greater clarity and emphasis?

**Do you want your identity to be public for this peer review?** For information about this choice, including consent withdrawal, please see our Privacy Policy

Reviewer #1: No

Reviewer #2: No

Reviewer #3: No

Reviewer #4: No

---

## [Author Response · Author response to Decision Letter 1]

17 Feb 2025

Emily Chenette,

Editor-in-Chief,

PlosOne.

February 7th, 2025

Subject: Rebuttal (Manuscript Reference Number PONE-D-24-42164)

Dear Professor Emily Chenette,

Enclosed is a revised manuscript by Ruinelli et al. entitled: “Usability of Machine Learning Algorithms Based on Electronic Health Records for the Prediction of Acute Kidney Injury and Transition to Acute Kidney Disease: A Proof of Concept Study."

We thank you and the reviewers for the time spent reviewing the paper. We would like to resubmit a corrected version of the manuscript that incorporates the comments and observations.

While the original meaning of the manuscript remains unchanged, the new version (attached is a copy showing deletions highlighted in blue and strikethrough font, additions in red font, and a clean version) has been refined. Also, please note that some of the pictures were edited due to the data analysis pipeline refinement suggested by reviewer #3 (highlighted in the text).

Below, you will find a point-by-point response to the reviewers' comments.

We are open to any additional comments or questions you may have.

Sincerely,

Antonio Bellasi,

---

## [Decision Letter · Decision Letter 1]

Dear Dr. Bellasi,

**Please provide minor revision requested, in particular:**
**-Please try to describe a hypothetical sample size in order to indicate the statistical power of the ML results and their associated NPV, particularly given the low incidence of AKI and AKD.**
**- Please address the points from Reviewer 3, in addition to further points from Reviewer 4 and if not possible include these limitations in the discussion.**

We look forward to receiving your revised manuscript.

Kind regards,

Fabio Sallustio, PhD

Academic Editor

PLOS ONE

**Journal Requirements:**

Reviewers' comments:

Reviewer's Responses to Questions

**Comments to the Author**

Reviewer #1: All comments have been addressed

Reviewer #2: All comments have been addressed

Reviewer #3: (No Response)

Reviewer #4: All comments have been addressed

2. Is the manuscript technically sound, and do the data support the conclusions?

Reviewer #1: Yes

Reviewer #2: Yes

Reviewer #3: No

Reviewer #4: Yes

3. Has the statistical analysis been performed appropriately and rigorously?

Reviewer #1: Yes

Reviewer #2: Yes

Reviewer #3: No

Reviewer #4: Yes

4. Have the authors made all data underlying the findings in their manuscript fully available?

Reviewer #1: Yes

Reviewer #2: Yes

Reviewer #3: Yes

Reviewer #4: Yes

5. Is the manuscript presented in an intelligible fashion and written in standard English?

Reviewer #1: Yes

Reviewer #2: Yes

Reviewer #3: Yes

Reviewer #4: Yes

**Reviewer #1:**  The authors performed the changes. They answered all the questions and modified the manuscript accordingly

I have nothing to add. Good luck!

**Reviewer #2:**  Dear Authors,

I found your work interesting from the start, but I believe it has been significantly improved thanks to the revisions received and your subsequent modifications. I think this has been an excellent team effort. I have no further comments.

**Reviewer #3:**  The revision form of the study “usability of machine learning algorithms for the prediction of AKI and transition to AKD” of Ruinelli and co-authors still now hasn’t answered to the points previously underlined.

Mainly three aspects appear to be limiting.

The first is concerning the mathematical process of machine learning.

The external validation consists on the testing of model in a different “external” group of patients, but in this case patients of external validation are those “internal” of hospital database. Therefore there isn’t an external validation.

The external test is necessary to prove the model performance and its ability to generalize on different geographical populations. (Ref. Alfieri et al, PLoS ONE 18(7): e0287398. https://doi.org/ 10.1371/journal.pone.0287398).

The second point is the calibration test. When a curve is generated, a Brier Score can be computed, and a Bias detection can be made. This aspect is not treated in the present study.

The third point is the very low Positive Predictive Value.

All these aspects deeply limit the study.

**Reviewer #4:**  Some other comments

- Please verify the font size, as it appears inconsistent across the first three lines.

- Was septic state considered in the model?

- I still believe it is essential to describe a hypothetical sample size in order to indicate the statistical power of the ML results and their associated NPV, particularly given the low incidence of AKI and AKD, as mentioned by the authors.

- A possible consideration, in addition to the absence of concomitant therapies, could be the potential role of gender in the progression to AKI or AKD. Are the tools included in the model represented or describe even in supplementary?

- As pointed out by the authors, the generalizability of the results remains debatable due to the lack of an external validation system for the ML model, and the cohort bias may be a significant factor influencing the results (low prevalence of AKI and AKD).

Despite these considerations, the work provides valuable insights and is well-structured and rigorous.

**Do you want your identity to be public for this peer review?** For information about this choice, including consent withdrawal, please see our Privacy Policy

Reviewer #1: **Yes: ** Lazar Chisavu

Reviewer #2: No

Reviewer #3: No

Reviewer #4: No

---

## [Author Response · Author response to Decision Letter 2]

16 May 2025

Emily Chenette,

Editor-in-Chief,

PlosOne.

May 14th, 2025

Subject: Rebuttal (Manuscript Reference Number PONE-D-24-42164)

Dear Professor Emily Chenette,

Enclosed is a revised manuscript by Ruinelli et al. entitled: “Usability of Machine Learning Algorithms Based on Electronic Health Records for the Prediction of Acute Kidney Injury and Transition to Acute Kidney Disease: A Proof of Concept Study."

We thank you and the reviewers for the time spent reviewing the paper. We would like to resubmit a corrected version of the manuscript that incorporates the comments and observations.

While the original meaning of the manuscript remains unchanged, the new version (attached is a copy showing deletions highlighted in blue and strikethrough font, additions in red font, and a clean version) has been refined. Also, please note that some of the pictures were edited due to the data analysis pipeline refinement suggested by reviewer #3 (highlighted in the text).

Attached, you will find a point-by-point response to the reviewers' comments.

We are open to any additional comments or questions you may have.

Sincerely,

Antonio Bellasi,

---

## [Editor Report · Decision Letter 2]

Usability of Machine Learning Algorithms based on Electronic Health Records for the Prediction of Acute Kidney Injury and Transition to Acute Kidney Disease: a Proof of Concept Study

PONE-D-24-42164R2

Dear Dr. Bellasi,

We’re pleased to inform you that your manuscript has been judged scientifically suitable for publication and will be formally accepted for publication once it meets all outstanding technical requirements.

Kind regards,

Fabio Sallustio, PhD

Academic Editor

PLOS ONE
---

## [Editor Report · Acceptance letter]

PONE-D-24-42164R2

PLOS ONE

Dear Dr. Bellasi,

I'm pleased to inform you that your manuscript has been deemed suitable for publication in PLOS ONE. Congratulations! Your manuscript is now being handed over to our production team.

Kind regards,

on behalf of

Prof. Fabio Sallustio

Academic Editor

PLOS ONE